# Clinical Impact of Sarcopenia on Gastrointestinal Tumors

Ana Pereira [1], Joaquim Costa Pereira [2] and Sandra F. Martins [2,3,4,*]

1   Surgery Resident, Braga Hospital, 4710-243 Braga, Portugal; anamariafppereira@gmail.com
2   Coloproctology Unit, Surgery Department, Braga Hospital, 4710-243 Braga, Portugal; jmcpereira62@gmail.com
3   School of Medicine, University of Minho, 4710-070 Braga, Portugal
4   PT Government Associate Laboratory, Life and Health Sciences Research Institute (ICVS)/3B's, 4710-070 Braga/Guimarães, Portugal
*   Correspondence: sandramartins@med.uminho.pt

**Abstract:** Preoperative risk stratification in cancer surgery is important to improve treatment and outcome. Sarcopenia is defined by progressive and generalized loss of skeletal muscle mass and strength and is now getting attention as a poor prognostic factor. The purpose of this review was to explore the impact of sarcopenia on short and long-term outcomes in patients undergoing surgical resection of gastrointestinal tumors. Recent studies suggest that sarcopenia contributes to postoperative complications and overall survival. The relatively simple evaluability, as well as its modifiable nature, provides an intriguing potential for sarcopenia to be included in standard preoperative clinical evaluation. Such evaluations can provide physicians with important information to target high-risk individuals with prophylactic measures and eventually improve surgical outcomes.

**Keywords:** gastrointestinal tumors; morbimortality; sarcopenia

## 1. Introduction

Advances in surgical techniques and developments in perioperative care have improved surgical outcomes in gastrointestinal cancer treatment [1–3]. However, the prognostic gain of tumor resection should be balanced against the substantial risk of adverse reactions such as anastomotic leakage, stenosis and infection. These postoperative complications can have significant implications in mortality, disease recurrence and tolerance to adjuvant therapies [4,5].

With the ageing of the world population, the number of elderly people with cancer is also growing. Older patients have more risk of developing perioperative complications and have higher mortality rates. Therefore, clinical assessment to determine individuals' complication risk is important to improve outcomes after cancer surgery. Albumin levels, American Society of Anesthesiologists (ASA) classification and emergency surgery are known to predict short-term outcomes, whereas advance age and disseminated disease determine long-term [6,7]. Nevertheless, outcomes of patients with similar characteristics (age, ASA classification or tumor stage) can be very diverse in clinical practice. This means that the risk factors generally used to predict outcomes after cancer surgery, may reflect the patient's general condition and physiological reserve insufficiently. An important risk factor for a worse outcome is frailty, which is a state of increased vulnerability toward stressors in older individuals, leading to an increased risk of developing adverse health outcomes. A hallmark sign of frailty is sarcopenia [8–12].

Sarcopenia was traditionally characterized as a deterioration in muscle mass that frequently happens to older people [13]. However, the European Working Group on Sarcopenia in Older People (EWGSOP) recommended that the sarcopenic phenotype should also include reduced muscular strength as well as physical function [14]. It can exist in normal weight or overweight patients, and that is why sarcopenia is not the same as weight loss or cachexia. Recent studies suggest that sarcopenia is independently associated

with poor prognosis in the treatment of gastrointestinal cancer. However, in the surgical field, sarcopenia did not receive the same attention and the assessment of muscle mass is not included in the usual perioperative evaluation [15,16].

With this background, we made this present review of the literature to explore whether preoperative incidence of sarcopenia is a predictor for postoperative complication risk after gastrointestinal cancer surgery.

## 2. Sarcopenia Definition and Assessment Methods

Sarcopenia can be defined by the progressive loss of skeletal muscle mass and function (strength and/or physical performance), leading to a bigger risk of physical incapacity, worse quality of life and death [14,17].

Rosenberg first supported the term "sarcopenia" in 1989, stating the importance of skeletal muscle mass reduction in older people [18]. In healthy individuals, the prevalence of sarcopenia growths with ageing, oscillating from 9% at 45 years and up to 64% in individuals with 85 years old or more [19]. Sarcopenia can be divided into primary or secondary. Primary sarcopenia is caused by ageing, whereas secondary is triggered by illness, malnutrition, invasive procedures, organ failure, cancer and other diseases [14]. An aggressive tumor biology with amplified metabolic activity, leads to systemic inflammation and muscle wasting. This can elucidate why skeletal muscle reduction is a poor prognostic factor. As ageing of the population increases worldwide, cases of primary sarcopenia, along with secondary sarcopenia due to malignancy, will inevitably growth.

Regarding the assessment of sarcopenia, in 2010 the European Working Group on Sarcopenia in Older People (EWGSOP) proposed diagnostic algorithms [14]. In 2014, the Asian Working Group for Sarcopenia (AWGS) also made some recommendations [20]. Essentially, sarcopenia is a concept for the elderly, so both the EWGSOP and AWGS classifications are related to people over 60 or 65 years of age.

The reduction of skeletal muscle mass can be assessed by the computed tomography (CT), magnetic resonance imaging (MRI), dual energy x-ray absorptiometry (DXA) or bioelectrical impedance analysis. The deterioration of muscle strength can be evaluated with the knee extension/flexion or handgrip strength. On the other hand, the decline in physical performance can be measured by the short physical performance battery test, the timed get-up-and-go test, gait speed or the stair climb power test [14]. An important limitation is that there are no consensual cut-off points for sarcopenia definition in the literature and they also depend upon the evaluation technique selected. Nevertheless, the EWGSOP recommends as cut-off points two standard deviations below the mean reference value [14]. As CT is largely used for clinical tumor staging, it turns out to be an accessible method for the diagnosis of sarcopenia. Using a single abdominal cross-sectional CT image, at the third lumbar vertebrae (L3), it can be projected the total body mass by assessing the skeletal muscle index ($cm^2/m^2$), that is calculated by the sum of skeletal muscle areas, at L3 level, and normalized for stature [21–24].

## 3. Prevalence of Sarcopenia

The prevalence of sarcopenia in patients undergoing surgery for gastrointestinal cancers (evaluated by the measurement of skeletal muscle mass by CT) was reported in some studies (see Table 1). Despite similar age and sex distribution between studies, there was an extensive variation in the prevalence of this condition.

Cohorts of patients with esophageal and gastric cancer reported a widespread prevalence of sarcopenia before surgery, ranging from 43% to 79% [25,26]. Less discrepancy in the prevalence of sarcopenia was detected among patients undergoing surgical resection of hepatocellular carcinoma (40.5–54.1%) [27,28], colorectal cancer (38.9–47.7%) [29,30] and hepatic colorectal metastases (17.0–19.4%) [31,32]. These discrepancies can be explained by the distinct diagnostic criteria used in different studies. Although these criteria had been recognized long ago, consensual standards are still not established. Therefore, different studies use different cut-off points for muscle mass and muscle strength.

**Table 1.** Prevalence of sarcopenia in gastrointestinal tumors.

| Reference | Malignancy | Prevalence (%) |
|---|---|---|
| Awad et al. [25] | Oesophageal and gastric cancer | Before NACRT: 57 Before resection: 79 |
| Yip et al. [26] | Oesophageal cancer | Before NACRT: 26 Before resection: 43 |
| Voron et al. [27] | Hepatocellular carcinoma | 54.1 |
| Itoh et al. [28] | Hepatocellular carcinoma | 40.5 |
| Lieffers et al. [29] | Colorectal cancer | 38.9 |
| Reisinger et al. [30] | Colorectal cancer | 47.7 |
| Van Vledder et al. [31] | Colorectal liver metastases | 19.4 |
| Peng et al. [32] | Colorectal liver metastases | 17 |

NACRT: neoadjuvant chemotherapy.

## 4. Sarcopenic Obesity

Sarcopenic obesity is a condition defined as obesity caused by age-induced muscle loss. As the muscle decreases, the body reduce its ability to burn fat, resulting in progressive obesity [33].

So far, the definition of sarcopenic obesity is not clear. Some define the sarcopenia group with a high body mass index (BMI) as sarcopenic obesity [15], while others use the visceral fat mass measure by CT to define it [34]. However, a large proportion of obese patients with cancer are affected by sarcopenia. The simple observance of body weight is insufficient to detect this abnormality; specific quantification of skeletal muscle is needed.

In cancer patients, the prevalence of sarcopenic obesity may be increasing. Comparable to cancer, sarcopenia is considerably prevalent in older people and the capacity of malignant disease to cause muscle atrophy increases this prevalence. Similarly, obesity is a recognized risk factor for some types of cancer, such as colon cancer [35].

Notably, it has been stated that sarcopenic obesity is associated with an increased risk of surgical complications and decreased survival [34]. This condition combines the health risks of obesity and depleted lean mass.

## 5. Relation between Sarcopenia and Surgical Outcomes

### 5.1. Colorectal Cancer and Hepatic Colorectal Metastases

An increased postoperative morbidity rate was found in patients with sarcopenia undergoing surgical resection of colorectal cancer and hepatic colorectal metastases. Furthermore, a cohort of 302 patients reported that an increase in psoas density protected against overall and infectious complications [36].

Nakanishi et al. found that sarcopenia has a significant association with higher incidence of all postoperative complications, especially for patients with Clavien–Dindo (CD) classification grade $\geq 2$. Among postoperative complications, sarcopenia seems significantly correlated with postoperative infections, especially those occurring in locations other than surgical sites—in other words, general infections [37]. Lieffers et al. reported that sarcopenia was an independent predictor for infections, nevertheless, they did not investigate specifics infections or other postoperative complications [29]. Another study presented a strong association between sarcopenia and 30-day mortality after elective colorectal cancer surgery in patients with and without sarcopenia (8.8% vs. 0%, respectively) [30].

In patients undergoing hepatic resection for colorectal metastases, it was described a higher risk of postoperative complications (CD grade IIIa or higher) in individuals with sarcopenia compared with those without [32].

*5.2. Gastric Cancer*

In what concerns to gastric cancer, Fukuda et al. have reported that serious postoperative complications (CD ≥ IIIa) are significantly more common in the sarcopenia group, showing that sarcopenia can be an independent prognostic factor for serious postoperative complications [38]. Huang et al. have emphasized the importance of sarcopenia stages defined by the EWGS [39]. They demonstrated that patients had worse postoperative outcomes following gastric surgery when they are in advanced sarcopenia stage. Zhuang et al. have similarly reported that sarcopenia is an independent prognostic factor for severe complications following surgery [40].

*5.3. Others*

Studies did not find an association between sarcopenia and postoperative morbidity and mortality in patients undergoing resection for esophageal or hepatocellular cancer [25,27,41]. Furthermore, in a cohort of 557 patients undergoing pancreatic cancer resection, there was no difference in the rate of any postoperative complication or 30-day postoperative mortality [42].

Two studies [30,41] that reported on anastomotic leakage following surgical resection of colorectal and esophageal cancer did not demonstrate an association with sarcopenia.

## 6. Length of Intensive Care Unit and Hospital Stay

Two studies described that sarcopenia was a predictor of longer postoperative hospital stays [29,32]. Length of hospital stay was slightly prolonged in patients with sarcopenia undergoing resection with curative intent for hepatic colorectal metastases (6.6 versus 5.4 days; *p* = 0.03) [32]. The same conclusion was described by Lieffers et al. in colorectal cancer surgery—sarcopenia is associated with increased health service utilization, in the form of increased length of stay and requirements for prolonged rehabilitation care [29].

Peng et al. reported a prolonged Intensive Care Unit (ICU) admission (>2 days) for patients with sarcopenia undergoing surgery with curative intent for hepatic colorectal metastases compared with those without sarcopenia (15% versus 4% respectively; *p* = 0.004) [32].

On the other hand, Jones et al. did not find any association between length of hospital stay and sarcopenia [43]. Length of hospital stay did not differ significantly between patients with and those without sarcopenia in studies of pancreatic cancer [44] and esophageal and gastric cancer [25,26]. Awad et al. [25] did not find an association between sarcopenia and hospital length of stay in 47 patients with gastroesophageal cancer patients. Further prospective studies are required to confirm and expand the findings available to date.

## 7. Relation between Sarcopenia and Overall Survival

Most authors stated a substantial decrease in overall survival in patients with sarcopenia compared with those without sarcopenia. This effect was observed regardless of cancer site or tumor origin [27,31–38].

Among patients with hepatocellular carcinoma, Voron et al. [27] reported a median survival time of 52.3 and 70.3 months in patients with and without sarcopenia respectively. In a study of individuals who had pancreatic cancer surgery, Peng et al. reported that the 3-year survival rate was inferior in patients with sarcopenia than in those without [44]. Even with the multivariable analysis, sarcopenia continued independently associated with an increased risk of death at 3 years.

In colorectal cancer surgery, Nakanishi et al. reported that sarcopenia did not significantly correlate with the overall or recurrence-free survival [37]. Black et al. discovered an association, on Kaplan Meier method, between sarcopenia and shorter overall survival. However, the outcomes were no longer significant when analyzed in a multivariate Cox regression [42]. Reisinger et al. found a significant association between 30-day postoperative mortality and the presence of sarcopenia [30]. Concerning the long-term mortality, Jung et al. recently stated that sarcopenia was not significantly related with the short- or

long-term overall survival in their study with 229 colorectal cancer patients (based on sex-adjusted psoas muscle index values) [45].

In patients with esophageal cancer, a study reported a tendency towards reduced survival among those with sarcopenia [26]. On the other hand, in Sheetz et al. [41], overall survival was decreased in individuals who had esophageal cancer with sarcopenia and did not take neoadjuvant chemotherapy, although no significant association was reported between patients who did receive neoadjuvant chemotherapy. Tamandl et al. [46] verified that sarcopenia might be a prognostic factor of overall survival, in patients with esophageal or esophagogastric junctional cancer (along with the T factor and a positive surgical margin).

Zhuang et al. have reported that sarcopenia is an independent prognostic factor for overall survival after radical gastrectomy for gastric cancer [40]. Moreover, in another study, sarcopenia was associated with chemotherapy toxicity in gastric cancer patients in the perioperative setting, leading to early discontinuation of chemotherapy and dose reduction [47].

## 8. Relationship between Sarcopenia and Disease-Free Survival

Sabel et al. reported that sarcopenia decreased disease-free survival in patients with primary colorectal cancer [36]. Furthermore, this study also described a protective effect of high psoas muscle density. However, in another study [45], there was no significant difference in disease-free survival between patients with normal and low skeletal muscle mass.

In what concerns to patients with hepatic colorectal metastases, Van Vledder et al. [31] reported a median disease-free survival time of 8.7 months in patients with sarcopenia compared with 15.1 months in patients without sarcopenia ($p = 0.002$).

Studies show that in esophageal cancer, sarcopenia was associated with decreased disease-free survival in those who underwent surgical resection without getting neoadjuvant chemoradiotherapy independently of age, sex and tumor. Nevertheless, no association between sarcopenia and disease-free survival was detected in patients who underwent surgical resection following neoadjuvant chemoradiotherapy [26,41].

Zhuang et al. have reported that sarcopenia is an independent prognostic factor for disease-free survival in stage II/III of gastric cancer [40].

## 9. Quality of Life and Depression

In patients recently diagnosed with incurable cancer, sarcopenia is associated with worse quality of life and depression. Investigators reported that sarcopenia is associated with physical deterioration which will decreases functional ability and can damage the quality of life of patients with cancer [48]. Similarly, the connection between depressive symptoms and lack of appetite or reduced physical activity may offer a link between sarcopenia and depression in patients with cancer [48]. With the evaluation of advanced cancer patients for the presence of sarcopenia, using routinely performed CT scans, we should be able to recognize those at higher risk for decreased physical and emotional well-being. Consequently, we can use strategies and different approaches to better answer the needs of these patients.

## 10. Pharmacological Interventions for Treatment of Sarcopenia

The causes of sarcopenia are not fully understood. An imbalance between muscle protein synthesis and degradation may cause the beginning of sarcopenia and various mechanisms are involved. Understanding these mechanisms is crucial to recognize molecular targets for pharmacological treatment of sarcopenia.

### 10.1. Myostatin

The most studied molecular target for muscle-wasting disease is myostatin. It is a member of the transforming growth factor ß (TGF-ß) superfamily and can also be called growth differentiation factor-8 (GDF-8). Myostatin is specially expressed in skeletal muscle

cells and inhibits myogenesis: muscle cell growth and differentiation [49]. Consequently, myostatin may be a good target in muscle-wasting diseases. It has been proposed as a primary method for pharmacological interventions. The first human trial tested the Stamulumab (MYO-029) which is a recombinant human antibody that inhibits the activity of myostatin protein [50].

### 10.2. Activin Receptor

Along with ligands such as myostatin and activins, the receptor ACVR2B has also been considered as a target for the development of drugs for sarcopenia and muscle-wasting disease. An ACVR2B anti-body neutralizes the signaling pathway and was described to induce muscle hypertrophy [51].

At this time, most of the compounds in progress have had limited efficacy in large clinical trials. Even though, many smaller clinical trials have confirmed that the inhibition of myostatin/ACVR2 signaling may improve muscle mass in patients with muscle-wasting disease. Some studies reported that exercise training and nutritional supple mentation have positive effects on muscle wasting. Therefore, combined therapies with myostatin inhibitors and other approaches such as exercise and nutritional therapy could be more effective in the treatment of sarcopenia [52,53].

### 10.3. Exercise Mimetics

Several studies have described the positive contribution of exercise on patients with sarcopenia. Exercise has consistently demonstrated improved muscle strength and function, with inconsistent effects on muscle mass. However, the majority of patients with sarcopenia have many limitations with physical activity. Consequently, exercise mimetics are a potential therapeutic strategy, because these pills produce the effects of exercise without exercise. AMP-activated protein kinase (AMPK) agonists are exercise mimetics but they are still in pre-clinical stages as a result of side effects [54].

### 10.4. Hormones

With aging, there is a decrease in circulating levels of some anabolic hormones. This contributes to changes in muscle mass and function in the elderly. Then, hormonal manipulation is another target of many of the therapies for muscle-wasting disease. In phase 2 of clinical trials, treatment with selective androgen receptor modulator (SARM) enobosarm (also known as ostarine, MK-2866) made dose-dependent increases in total lean body mass with developments in physical function in older individuals. Despite these advances related with hormonal targets, it is still premature to evaluate the clinical efficiency of hormonal supplementation for the management of sarcopenia [55,56].

## 11. Discussion

Some conclusions can be drawn from this review about the influence of CT-assessed sarcopenia on short and long-term outcomes in resectable gastrointestinal malignancies. Sarcopenia decreased overall survival. Patients with sarcopenia undergoing surgery for colorectal cancer and hepatic colorectal metastases had a prolonged length of stay in hospital and an increased postoperative morbidity rate. In what concerns to gastric cancer, it seems that serious postoperative complications are significantly more common in the sarcopenia group.

Therefore, sarcopenia must have an important influence on early postoperative outcomes and prognosis. Patients with sarcopenia have physical deterioration and decreased functional ability, making them more vulnerable to the physiological stresses of surgery and more at risk of complications. As a matter of fact, the association between sarcopenia and frailty is well recognized. This raises the possibility that, as opposed to other preoperative prognostic indicators (e.g., TNM staging), sarcopenia could be modified via prehabilitation.

Preoperative risk stratification is of extreme importance in patient selection for surgery, as it may help physicians to recognize patients with a high risk of worse outcome after the

procedure. An instrument appropriate for risk evaluation should be inexpensive, easily available and reliable. Bioelectrical impedance analysis, dual-energy X-ray absorptiometry and skinfold measurement are often not performed routinely during the oncological evaluation, whereas the majority of patients undergo abdominal CT in the pre-operative investigations.

Cross-sectional muscle area can be measured quickly by single-slice examination of abdominal CT images, and is linearly associated to total body skeletal muscle mass [22]; this measurement has a short level of inter observer inconsistency [22,30]. In patients with cancer, the CT-based skeletal muscle mass measurement may recognize those at an initial stage of frailty, which would otherwise have been unnoticed clinically [57].

Even though the appearance of protocols like the enhanced recovery after surgery (ERAS) reduces postoperative weight loss, focusing on the prevention of muscle mass loss might be more significant than simple weight control. For example, individualized physical activity delivered in the period between diagnosis and treatment has been shown to be feasible, safe and reduce postoperative complications in patients with colorectal cancer [45]. Beyond that, a recent systematic review has reported that the positive outcomes of prehabilitation can be seen after just 2 weeks [45]. Consequently, the pretreatment assessment of sarcopenia could be an effective and powerful way of choosing patients who would take more benefits from such targeted and individualized prehabilitation.

We still do not know if pharmacological interventions on sarcopenia can improve outcomes. Nevertheless, understanding the mechanism of sarcopenia is important for the development of therapeutic strategies. Studies about muscle wasting in cancer has significantly increased over the past decade [52] and has led to new treatment possibilities, such as myostatin inhibitors [58,59]. However, the association of non-drug therapies is also crucial. Exercise and nutritional supplements are more accessible as intervention strategies against sarcopenia instead of pharmacological treatments alone. Currently, there are several clinical trials investigating the stabilization or reversal of muscle wasting in patients with cancer [60].

This review has some limitations. The included studies were heterogeneous in design and the vast majority were retrospective. Furthermore, there is no standard definition of CT-based assessment of muscle mass and different methods were used, which also is a disadvantage in the evaluation of the results.

## 12. Conclusions

During the preoperative evaluation of patients with cancer, the measurement of muscle mass is an easy and inexpensive diagnostic method of sarcopenia, since it is also done for staging. In the presence of this condition, we can predict a higher possibility of worse surgical outcomes and, therefore, we can interfere in an opportune and prophylactic way.

It is documented that sarcopenia reduces overall survival and may increase postoperative morbidity. The measurement of muscle mass by CT may contribute in preoperative decision-making, mainly in patients considered to be incapable for surgery or who have a high risk of complications. Those sarcopenic patients could then go through targeted strategies such as prehabilitation, to upgrade both their short and long-term outcomes.

**Author Contributions:** Conceptualization, A.P., J.C.P. and S.F.M.; methodology, A.P., J.C.P. and S.F.M.; validation, A.P. and S.F.M.; formal analysis, A.P.; investigation, A.P.; writing—original-draft preparation, A.P.; writing—review and editing, A.P., J.C.P. and S.F.M.; supervision, J.C.P. and S.F.M.; project administration, S.F.M. All authors have read and agreed to the published version of the manuscript.

**Funding:** This research received no external funding.

**Conflicts of Interest:** The authors declare no conflict of interest.

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
