# Peer review of "Clinical Impact of Sarcopenia on Gastrointestinal Tumors"

_gastrointestdisord, doi:10.3390/gidisord3010006_

Round 1

Reviewer 1 Report

I appreciate the opportunity to review this paper on the clinical impact of sarcopenia on gastrointestinal tumours.

 This paper's genesis is an observation that sarcopenia is independently associated with poor prognosis in cancer management. In their paper, the authors have reviewed the literature to explore whether sarcopenia's preoperative incidence is a predictor for postoperative complication risk after gastrointestinal cancer surgery.

Recently, many reviews and meta-analyses have appeared regarding sarcopenia on a similar but not the same topic, and the studies analysed by the authors overlap with those analysed in previous publications [1-6].

I enjoyed reading the manuscript. I commend the authors for several strengths of their work, including clinically relevant study question.

The authors conducted a comprehensive literature search (two different authors conducted the search for previous studies separately). There were clear eligibility criteria for studies being chosen or rejected for the review, and the authors provided the characteristics of the studies listed. The authors also provide a satisfactory discussion of some heterogeneity observed in the results of the review.

Considering these strengths, though, as I read the manuscript, I found some areas where I would have appreciated greater clarity.

  • The manuscript title should Identify the paper as a review.
  • Authors should describe more clearly their rationale for the review in the context of the existing knowledge.
  • It would be beneficial to readers if the authors give numbers of studies screened, assessed for eligibility, and included in the review, with reasons for exclusions at each stage, with a flow diagram.
  • The current definition of sarcopenia, unlike the previous ones, pays more attention to the reduction of muscle strength and physical impairment than just to the loss of muscle mass [7].
  • The authors should, even more, pay attention to this fact in the discussion. Only considering muscle mass can lead to false conclusions. Please provide a general interpretation of the review results and implications for future research.
  • When discussing the study's limitations, the authors should also consider the incomplete retrieval of identified research. I understand they limited their search to only one database. Have authors conducted a manual search through references of articles?
  1. Pipek, L.Z.; Baptista, C.G.; Nascimento, R.F.V.; Taba, J.V.; Suzuki, M.O.; do Nascimento, F.S.; Martines, D.R.; Nii, F.; Iuamoto, L.R.; Carneiro-D'Albuquerque, L.A. The impact of properly diagnosed sarcopenia on postoperative outcomes after gastrointestinal surgery: A systematic review and meta-analysis. PloS one 2020, 15, e0237740.
  2. Su, H.; Ruan, J.; Chen, T.; Lin, E.; Shi, L. CT-assessed sarcopenia is a predictive factor for both long-term and short-term outcomes in gastrointestinal oncology patients: a systematic review and meta-analysis. Cancer Imaging 2019, 19, 1-15.
  3. Kudou, K.; Saeki, H.; Nakashima, Y.; Edahiro, K.; Korehisa, S.; Taniguchi, D.; Tsutsumi, R.; Nishimura, S.; Nakaji, Y.; Akiyama, S. Prognostic significance of sarcopenia in patients with esophagogastric junction cancer or upper gastric cancer. Annals of surgical oncology 2017, 24, 1804-1810.
  4. Simonsen, C.; de Heer, P.; Bjerre, E.D.; Suetta, C.; Hojman, P.; Pedersen, B.K.; Svendsen, L.B.; Christensen, J.F. Sarcopenia and postoperative complication risk in gastrointestinal surgical oncology: a meta-analysis. Annals of surgery 2018, 268, 58-69.
  5. Lukovich, P.; Nagy, Á.; Barok, B.; Csiba, B.; Rokka, R.; PÅ‘cze, B. Nutritional Status, Sarcopenia and the Importance of Prehabilitation of Gastrointestinal Tumor Patients: a Surgical Point of View. Journal of Nutritional Oncology 2020, 5, 132.
  6. Song, H.; Dong, M. Sarcopenia as a novel prognostic factor in the patients of primary localised gastrointestinal stromal tumor. 2020.
  7. Cruz-Jentoft, A.J.; Sayer, A.A. Sarcopenia. The Lancet 2019, 393, 2636-2646.

Author Response

Thank you very much for your suggestions. We send the list of related changes that are highlighted in the text

Revisor 1

1 - The manuscript title should Identify the paper as a review

Answer: the title was revised, and this information was included

2- Authors should describe more clearly their rationale for the review in the context of the existing knowledge

Answer:  This information was revised and included in the introduction section

3- It would be beneficial to readers if the authors give numbers of studies screened, assessed for eligibility, and included in the review, with reasons for exclusions at each stage, with a flow diagram.

Answer: The methods section was revised. We included the eligibility criteria and the number of original studies included. However, we did not make a flow diagram.

4 - The current definition of sarcopenia, unlike the previous ones, pays more attention to the reduction of muscle strength and physical impairment than just to the loss of muscle mass [7].

Answer: Considering the paper about sarcopenia [7], we revised and completed the section about sarcopenia definition.

5 - The authors should, even more, pay attention to this fact in the discussion. Only considering muscle mass can lead to false conclusions. Please provide a general interpretation of the review results and implications for future research.

Answer:  We revised the discussion section giving more attention to the reduction of muscle strength and physical impairment in the definition of sarcopenia.

6 - When discussing the study's limitations, the authors should also consider the incomplete retrieval of identified research. I understand they limited their search to only one database. Have authors conducted a manual search through references of articles?

Answer:  Yes, we limited our study to only one database and we did not conducted a manual search. The study limitations were revised and this information was included.

Reviewer 2 Report

This review focused on the clinical impact of sarcopenia on the surgical and oncologic outcome after resection of gastrointestinal tumors (cancers).

The manuscript is very well written and structured, describing the definitions, assessement, incidence, surgical and oncologic outcomes, and intervention as well.

MINOR POINTS:

1) 1. Introduction, the last paragraph: The authors stated the purpose of this review was to explore the impact on sarcopenia on short- and long-term outcome in the Abstract; however, in this paragraph the authors was concerned whether sarcopenia was a predictor of postoperative complication risk.

2) 5. Conclusions, the second paragraph:  Sarcopenia impairs overall survival.... , yes,  in some studies, as well as disease-free survival. There were some studies that failed to demonstrate the association between sarcopenia and prognosis, and the majority of the included studies were retrospective. Therefore, it might be too strong to conclude 'sarcopenia impairs overall survival'. Please consider replace the sentence with more modest expression.

Author Response

Thank you very much for your suggestions. We send the list of related changes that are highlighted in the text

Revisor 2

1 - Introduction, the last paragraph: The authors stated the purpose of this review was to explore the impact on sarcopenia on short- and long-term outcome in the Abstract; however, in this paragraph the authors was concerned whether sarcopenia was a predictor of postoperative complication risk.

Answer: The abstract was revised.

2 - Conclusions, the second paragraph:  Sarcopenia impairs overall survival.... , yes,  in some studies, as well as disease-free survival. There were some studies that failed to demonstrate the association between sarcopenia and prognosis, and the majority of the included studies were retrospective. Therefore, it might be too strong to conclude 'sarcopenia impairs overall survival'. Please consider replace the sentence with more modest expression.

Answer:  The conclusion was revised.